# Technical Validation of Ultrasound Assessment of the Thyroid Gland in Cattle

**DOI:** 10.3390/vetsci10050322

**Published:** 2023-04-28

**Authors:** Justine Eppe, Patrick Petrossians, Valeria Busoni, Frédéric Rollin, Hugues Guyot

**Affiliations:** 1Clinical Department of Production Animals, Fundamental and Applied Research for Animals & Health Research Unit (FARAH), Faculty of Veterinary Medicine, University of Liège, Quartier Vallée 2, Avenue de Cureghem 7A–7D, 4000 Liege, Belgium; frollin@uliege.be (F.R.);; 2Department of Endocrinology, University Hospital of Liege, University of Liege, Quartier Hôpital, Avenue Hippocrate 13, 4000 Liege, Belgium; patrick.petrossians@chuliege.be; 3Department of Clinical Sciences of Equids, Equine Division, Diagnostic Imaging Section, Fundamental and Applied Research for Animals & Health Research Unit (FARAH), University of Liege, Quartier Vallée 2, Avenue de Cureghem 5, 4000 Liege, Belgium; vbusoni@uliege.be

**Keywords:** cattle, ultrasonography, diagnostic imaging, metabolism, endocrine system

## Abstract

**Simple Summary:**

Cattle are not routinely screened for thyroid anomalies. Scientific data regarding thyroid diseases are also scarce. This may be due to the lack of easily accessible tools to assess the thyroid function of these animals. Thyroid ultrasound examination is a cheap, easy and non-invasive technique that can be performed without sedation. Apart from morphological information, it may also provide functional data about this gland. In this study, we attempted to validate the use of thyroid ultrasound in cattle by measuring intra- and inter-observer variability. Thyroid estimated measurements were repeated in five calves and five cows, by three different operators with different trainings. Results show that this examination has good repeatability and may be used more routinely in veterinary practice.

**Abstract:**

Little is known about thyroid diseases in ruminants, probably due to the lack of diagnosis techniques developed in this species. However, thyroid ultrasound (TU) is widely used in human and in companion animal’s medicine. It is a cheap and non-invasive examination, which allows for the identification of thyroid structures or diffuse diseases. The aim of this study was to evaluate the accuracy of TU in five calves and five cows through inter- and intra-observer repeatability. The size of the thyroid gland was measured from three views: left sagittal, right sagittal and transverse; nine measurements per view. The intra-observer coefficient was calculated for each observer. For the inter-observer, the first observer was a board-certified imagist (European College of Veterinary Diagnostic Imaging diplomate), the second was a board-certified specialist in bovine and herd management (European College of Bovine Health Managementdiplomate) and the third was an in-trained veterinarian for the TU. They each scanned the thyroid gland successively, following the same method. The intra-observer variabilities for observers 1, 2 and 3 were 8.22%, 5.53%, 5.38%, and 7.18%, 8.65% and 6.36%, respectively, for calves and cows. The inter-observer variability for calves was 10.4% and for cows, 11.8%. This study confirms the feasibility of repeatable intra- and inter-observer TU-estimated measurements in cattle.

## 1. Introduction

The endocrine system of ruminants, and more specifically, their thyroid metabolism, are not routinely studied, although they may have implications in milk and meat production. Several studies have shown that thyroid metabolism plays an important role in regulating the (negative) energy balance [1,2], reproductive system [3,4,5], fetal development [1] and milk production [6,7,8] of cows. Although diseases related to dietary iodine intake have already been reported in ruminants, such as hypothyroidism [9,10,11,12], hyperthyroidism [9] or iodine intoxication [13,14], the scientific literature does not mention any other thyroid disease in this species. In addition, thyroid hormones are involved in other disorders, such as acute respiratory distress syndrome in calves [15].

Consequences of thyroid gland dysfunction are widely described in humans, and especially in women [16]. There are close links between thyroid metabolism and milk production in breastfeeding women [17,18]. Further, a dysfunction of the thyroid metabolism is associated with many complications of pregnancy in women, such for example fetal growth restriction, fetal distress syndrome, hypertension, postpartum hemorrhage, and many others [19].

The scarcity of scientific literature regarding thyroid diseases in ruminants could be explained by the subclinical aspect of thyroid diseases in cattle, combined with the lack of diagnostic tools. Therefore, it would be advisable to provide more detailed data on this metabolism in cattle by developing new diagnostic tests, since at present, only iodine nutritional markers and total thyroxine (T4) are routinely analyzed.

In human medicine, thyroid ultrasound (TU) is one of the most sensitive imaging examinations to evaluate the size of this gland and its structures (nodules, cysts) [20,21,22]. This examination also allows the detection of diffuse thyroid diseases, such as thyroiditis. TU is one of the ancillary examinations systematically carried out in humans following the detection of a thyroid disorder revealed by blood examination [19,21,22,23].

The same TU method is used in dogs and cats, with differences in size, shape and echogenicity of the thyroid depending on breed, size and age of the animal [24,25,26]. In dogs, ultrasound has a 98% sensitivity in detecting hypothyroidism when the thyroid volume measurement is combined with its TU pattern [27]. In horses, ultrasound is recognized as a reliable means of examining the neck region, including the thyroid [28,29].

In ruminants, two studies [30,31] have recently studied the ultrasonographic, anatomical examination and Computed Tomography Scanner appearance of normal thyroid in goats. One of these studies has concluded that the thyroid gland is clearly visible in all the goats in the study and identified a standard TU measurement for goats [30]. No significant difference was found between the two lobes.

In bovine medicine, a first study [32] was carried out to determine the normally visible ultrasound structures of the neck, with the aim of exploring masses and phlebitis. There is currently no standardized TU method for cattle, whereas this examination is inexpensive and easily practicable on a standing animal. Two recent studies have used TU in calves [33,34], using the same method described in human medicine [20,21,22,35]. It appears that TU is non-invasive, inexpensive and easy to perform on standing calves [33,34]. The study of Metzner et al. (2015) [34] reported a correlation between thyroid axis measurements, thyroid volume and animal weight. The coefficient of variation of the measurements was worse for thickness, but better for length and width [34]. However, no further literature on the subject is available, and much work remains to be done to validate this TU examination of the thyroid as reliable.

The aim of this study was to test the feasibility of a method of TU in calves and cows without sedation and to assess its inter- and intra-observer repeatability. Indeed, although TU has already been tested in calves [34], the repeatability of the measurements has never been measured, although this is necessary to validate a standardized method.

## 2. Materials and Methods

All procedures have been approved by the Ethics Committee of the University of Liege (file number: 2224).

### 2.1. Animals

The study population was composed of 5 calves and 5 cows. Five calves, four Holstein Friesian (HF) and one Belgian Blue (BB) cattle breed, four female (HF) and one male (BB), aged of 16 ± 9 days and weighed 60 ± 8 kg, and five cows, all Holstein Friesian aged of 4 ± 1 years, with a body condition score (BCS) of 3 ± 0.5 on a scale of 1 to 5 [36] and weighed 794 ± 34 kg were enrolled in this study. They all came from the experimental farm of the University of Liege. All animals were examined before performing the study and only clinically healthy animals were retained. The neck area was clipped. Alcohol was applied to the skin and ultrasound gel was used to perform the TU. Restraint was done manually by a third party for standing calves and cows, by means of a feed fence and a halter. Neither sedation nor anesthesia was required.

### 2.2. Observers

The first observer (OBS1) was a board-certified imagist of the European College of Veterinary Diagnostic Imaging (ECVDI diplomate), the second observer (OBS2) was a board-certified specialist of the European College of Bovine Health Management (ECBHM diplomate) and the third observer (OBS3) was an in-trained veterinarian for TU. The observers scanned the animals successively, following the same method.

### 2.3. Ultrasonography

An ultrasound scanner (Mindray, DP-50VET, Shenzhen, China) equipped with a linear probe (10 mHz) for the calves or a convex probe (5 mHz) for the cows was used.

The thyroid was observed from 3 views: left sagittal, right sagittal, transverse (Figure 1). For transverse images, the probe was placed below the cricoid cartilage of the larynx, perpendicular to it, on either side of the trachea. The thyroid was scanned craniocaudally before being frozen on the most representative image, i.e., the one where the lobe appears largest and the isthmus is clearly visible. In the transverse view, the trachea was observed in section surrounded by the two lobes of the thyroid connected by an isthmus. The isthmus should be visible to consider the view successful [34]. This view made it possible to estimate the maximal height and width of the thyroid and to appreciate its symmetry, with splitting the field of view of the ultrasound scanner. For the sagittal view, the probe was rotated 90° from the transverse view, parallel to the axis of the trachea, and the thyroid was scanned from the medial to lateral position. The sagittal images allowed estimates of the maximal longitudinal axes of the right and left lobes (length). The maximum thickness of the isthmus was also estimated. A total of 9 observations per parameter was made using the measurement function (caliper) of the ultrasound device. This represents 315 estimates per observer for each animal category (cow or calf). For each estimation, the instruction was to take an image where it was possible to take the widest estimated measure of the thyroid. The intra-observer coefficient of variation was calculated for each observer in this study.

### 2.4. Statistics

The statistical analysis was performed using the R statistical package (The R Project for Statistical Computing, version 4.2.1, Vienna, Austria). The averages of the 9 measurements made of length, width, thickness of each lobe and the thickness of the isthmus of animals were compared according to the lobe. The means of all widths, lengths and thicknesses for the left and right lobes, respectively, were subjected to a Shapiro test. Depending on the result, if the data followed a normal distribution, an unpaired t-test was performed. If the data did not follow a normal distribution, a Wilcoxon test was performed. Inter-observer estimated measurements were compared using a Bland–Altman plot. Intra-observer variability was calculated using the mean of all the coefficients of variation (CV) of the nine consecutive estimates performed by the observer on each variable of each animal. Inter-observer variability was calculated using the mean of all the CVs between the observers measured variables.

## 3. Results

### 3.1. Ultrasound Technique and Measurements

The thyroid was easily accessible for ultrasound examination in calves and cows (Figure 1). It appeared hyperechoic or isoechoic compared to adjacent tissues, such as the sternohyoid and sternothyroid muscles. The parenchyma of the gland was surrounded by a hyperechoic capsule, allowing it to be easily delineated. In the cow, the thyroid gland was deeper, so it was necessary to ensure that the entire thyroid gland was present before estimated measurements were made. If the cow was fat, penetration was reduced and the gland was less visible (Figure 2). Only 2/5 cows were very distinct in BCS, so we did not perform statistics on these two individuals alone to see if the estimations differences were significantly different or not.

The means of the estimated measurements achieved by the three observers and their standard deviations are available in Table 1. The estimations were compared between the right and left lobes. Only one estimation was significantly different between the left and right lobes: the width of the right lobe of the cows was significantly smaller (*p* value < 0.05) than that of the left lobe.

### 3.2. Intra- and Interobserver Variability

The intra-observer variabilities for OBS1, OBS2 and OBS3 were 8.22%, 5.53%, 5.38% and 7.18%, 8.65% and 6.36%, respectively, for calves and cows.

The inter-observer variability for calves was 10.4% and for cows, 11.8%. Differences between estimations are available in Figure 3 and Figure 4. In calves, the correlation between the estimated measurements of observers 2 and 3 was particularly strong. In cows, the Bland–Altman plots in Figure 4 were similar and there were more differences between the estimations compared to calves, resulting in larger inter-observer coefficients in cows.

## 4. Discussion

To the best of the authors’ knowledge, this is the first study of intra- and inter-observer variability in bovine TU and the first study to technically validate a TU technique in cattle.

In this study, an ultrasound technique similar to Metzner et al. (2015) [34] was used. Our findings show that TU is a fast (taking less than five minutes), convenient and safe examination to be performed on cattle. Indeed, as it does not require sedation and is non-invasive, it only needs the usual means of restraint (feed fence, halter).

The thyroid gland is only a few centimeters away from the skin surface and is easy to scan. The gland parenchyma is isoechoic or hyperechoic compared to adjacent tissues such as muscles and is surrounded by a hyperechoic capsule. These characteristics are similar to those described in calves and other species [25,29,32,34].

The estimated measurements obtained for the calves are close to what has been previously described [34], although breeds, average age and weight differed. Measurements of the height (left lobe [LL]: 10.4 ± 1.7 mm; right lobe [RL]: 9.7 ± 1.5 mm), length (LL: 30.1 ± 3.0; RL: 30 ± 3.6) and width (LL: 22 ± 2.7; RL: 22.1 ± 2.5) of the lobes are described [34]. The coefficients of variation of these different measurements were calculated and varied between 3.04% to 7.26%, the coefficients of variation of the measurements of height being the highest. For cows’ TU, there were no data in the literature to compare with our findings. When comparing measurements between the right and left lobes, there is only one significant difference in the width of the right lobe in cows, which is smaller than the left. Few differences between left and right lobes were observed in other studies in calves [34], dogs [25] and horses [29]. Another study did not notice any difference between the two lobes during TU on goats [30]. Although our population size is sufficient to assess intra-/inter-observer variation for TU, it is insufficient to make clear conclusions about normal ultrasound dimensions of the thyroid gland in cattle.

The inter- and intra-observer values that we measured are very close to those reported in human medicine. Indeed, a study in humans [37] reported 8.4 +/− 6.7% for intra-observer and 13.3 +/− 8.2% for inter-observer variability for TU in 30 healthy children. Another study in human medicine gives however higher values; another study with 3 observers obtained an intra-observer variability of 14%, 13%, 19% and an inter-observer variability of 6.5%, 13.12%, 19.31% on 28 healthy adults [38]. A study in five dogs, comparing the volume of the thyroids on a measurement of length, width and thickness of each lobe, showed relatively higher intra- and inter-observer variability than that found in human medicine, especially for the length of the lobes [39]. This last study is the one with the most comparable design to ours.

In our study, we also observed two very different estimated measurements on the Bland–Altman Plot between OBS 2 and 3 in calves (Figure 3c). One hypothesis for these differences is that the estimations were made by measuring the thyroid without taking the maximum axes each time (due to movements of animals, depth of the skin, etc.). OBS 2 and 3 have trained extensively in calves TU, as this organ is rarely scanned in current veterinary cattle imaging practice. It is likely that the experience gained from this long and specific training on this organ reduces the differences in TU estimated measurements. These phenomena are also observed in another study [38]. In comparison, the absence of this difference in training for cow TU may explain the similar correlations in cows between all observers.

Several precautions should be taken when performing TU, before taking the images: for the transverse view, one should make sure to be in an area with the maximum thickness of the isthmus and to be perpendicular to the trachea; for the sagittal view, one must be parallel to the jugular gutter and be careful to visualize the entire thyroid. The last parameter to pay attention to is the pressure on the probe, as it changes the shape of the thyroid. For this parameter, it is very difficult to give a method and to objectify. All these precautions make it possible to obtain as close as possible standardized measurement, and thus, to avoid relevant or significant differences between observers [39].

It is documented in human medicine that it is very complicated to interpret ultrasound images of obese patients, especially for deep organs, because the adipose tissue absorbs and distorts the ultrasound rays, causing a decrease in contrast and image sharpness [40,41]. In a study comparing several ultrasound probes in obese adults, a very low frequency probe (1–3.5 mHz) and high element density provided the best image [41]. In our results, in fat adult cows, the thyroid is more difficult to visualize, but this does not seem to prevent its measurement. The images are darker, but there is no loss of sharpness (Figure 2). The fact that the thyroid gland is close to the surface of the skin undoubtedly makes it possible to reduce this effect, though perhaps a modification of the adjustment of the probe, such as, for example, decreasing the frequency, could improve the image in some cases. A study on a larger number of animals would be necessary to isolate an effect of overweight status on TU measurements, and the possible differences that could be obtained because of this condition.

TU is a widely used examination in human medicine, and is combined with other diagnostic tests (such as hormone assays) to make the diagnosis of thyroid diseases [22,23]. It is also used in follow-up for patients with thyroid diseases. In dogs, ultrasound is a useful tool for the diagnosis of hypothyroidism [42].

This study validates thyroid ultrasound as a repeatable means of measuring the thyroid in cattle. However, it does not provide standards for measuring the thyroid gland in cattle. Further studies should focus on the correlation of thyroid volume measurements by comparing TU with post-mortem measurements. In addition to ultrasound, the blood thyroid status of the animals should be assessed in order to establish standards for TU measuring. Further, if lesions are observed during TU, they should be related to their postmortem appearance, histology and blood status of the animals.

## 5. Conclusions

Thyroid ultrasound in cattle is a safe, easy to perform, repeatable and non-invasive examination. This repeatable test could benefit from further study for use in estimating the thyroid gland volume in cattle.

## Figures and Tables

**Figure 1 vetsci-10-00322-f001:**
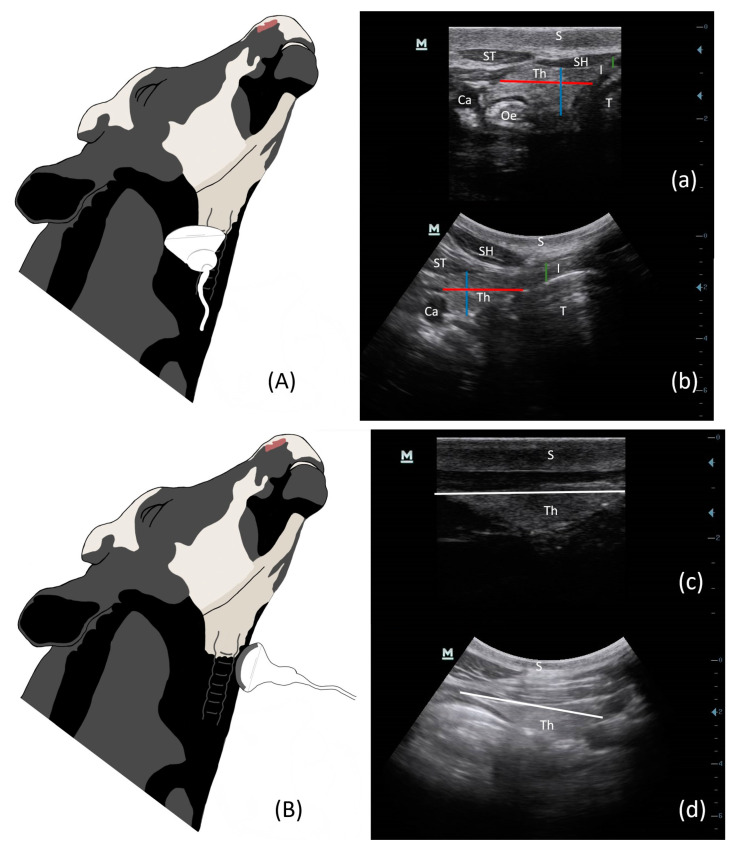
Illustration of TU in cows with a convex probe for transverse view (**A**) and sagittal view (**B**). TU images of the calf in left transverse (**a**), left sagittal (**c**) and of the cow in left transverse (**b**), left sagittal (**d**). The lines represent measurements taken on the lobe: red = width; blue = height; white: length. green = height of the isthmus. S = skin; Th = thyroid lobe; I = isthmus; Ca = carotid; Oe = oesophagus; T = trachea; ST = sternothyroid muscle; SH = sternohyoid muscle.

**Figure 2 vetsci-10-00322-f002:**
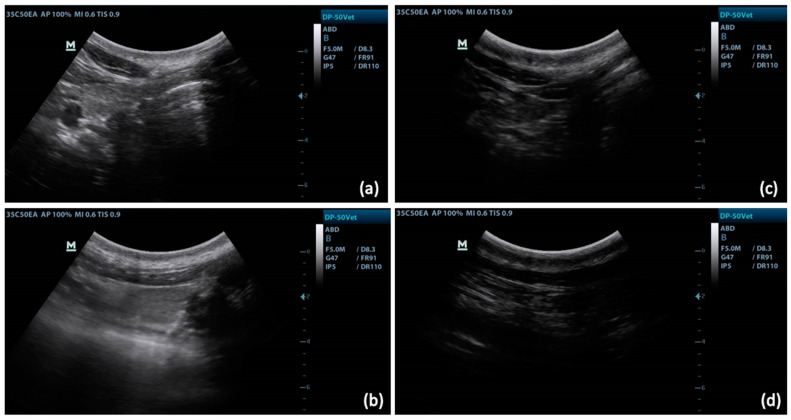
Comparison of thyroid images between a cow in normal or lean body condition (BCS 2.25/5; (**a**,**b**)) and a cow with a high or excessive body score (BCS 3.5/5; (**c**,**d**)). For the same ultrasound parameters (5 mHz; 47 of gain), the image is darker and it is more difficult to assess the boundaries of the gland in (**c**,**d**), compared to (**a**,**b**).

**Figure 3 vetsci-10-00322-f003:**
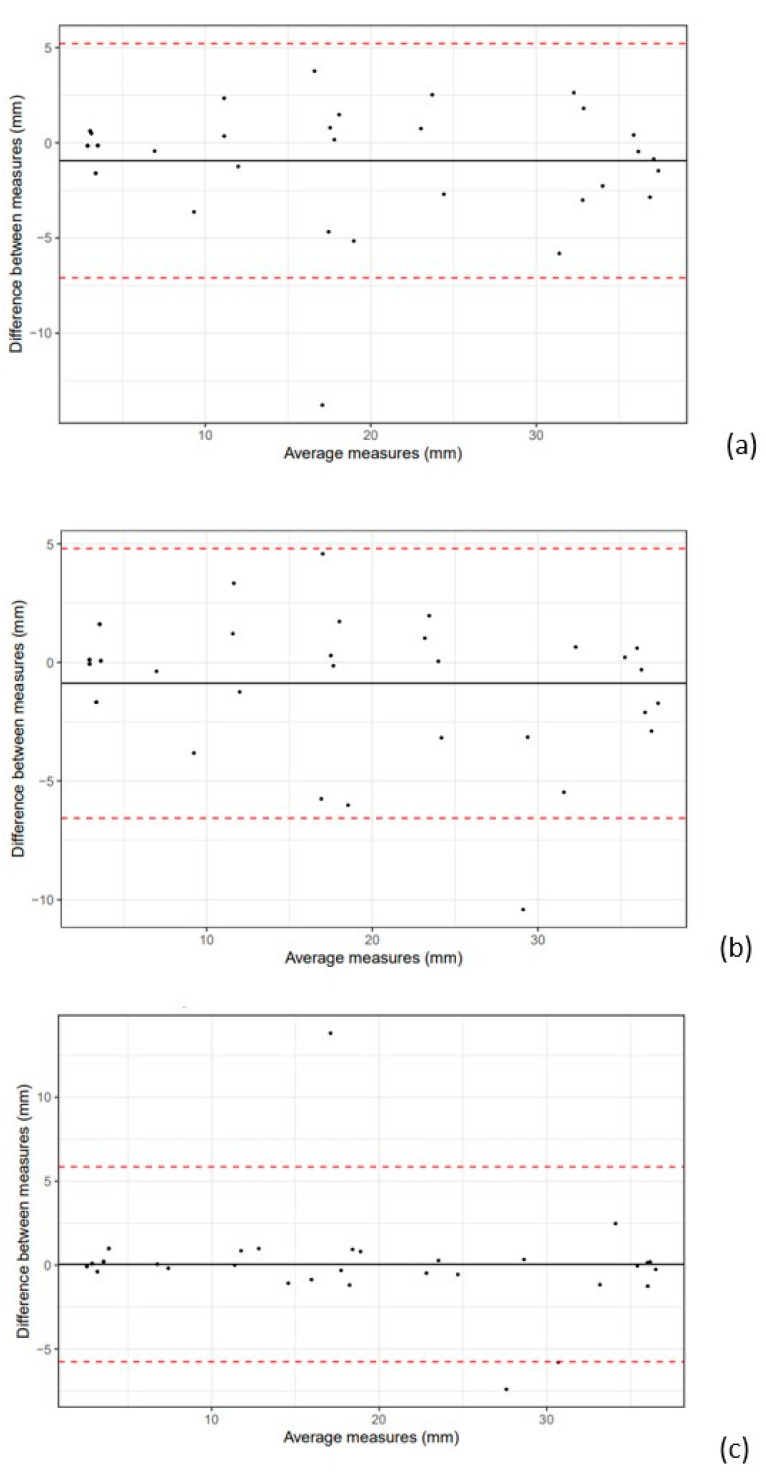
Bland–Altman plot in TU estimated measurements in calves between OBS 1 and 2 (**a**); OBS 1 and 3 (**b**); OBS 2 and 3 (**c**). Dashed lines represent the 95% confidence interval.

**Figure 4 vetsci-10-00322-f004:**
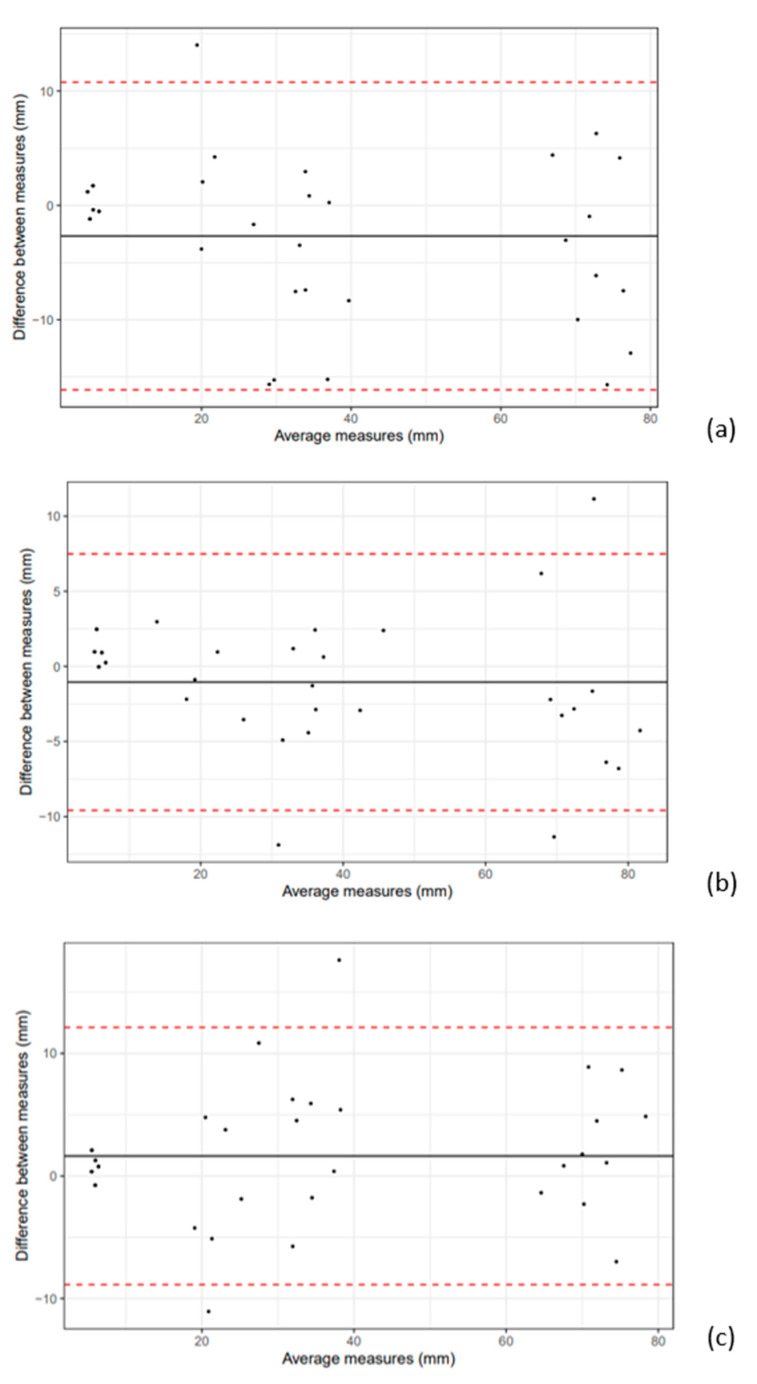
Bland–Altman plot in TU estimated measurements in cows between OBS 1 and 2 (**a**); OBS 1 and 3 (**b**); OBS 2 and 3 (**c**). Dashed lines represent the 95% confidence interval.

**Table 1 vetsci-10-00322-t001:** Means ± standard deviation of the estimates (mm) of the thyroid gland in calves and cows.

	Calves (n = 5)		Cows (n = 5)	
	LL	RL	LL	RL
**Height (mm)**	10.1 ± 2.4	8.8 ± 1.8	21 ± 4.6	22.8 ± 4.2
**Width (mm)**	20.4 ± 4.2	18.7 ± 3.2	37.2 ± 5.6	31.7 ± 5.5 *
**Length (mm)**	33.5 ± 3.8	34.6 ± 2.8	72.5 ± 6.7	72.9 ± 5.4
**Isthmus thickness (mm)**	3.2 ± 0.6		5.7 ± 1.04	

LL = left lobe, RL = right lobe. * Wilcoxon test significant results (*p* < 0.05).

## Data Availability

All data are available in the manuscript.

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
