# Peer review of "Technical Validation of Ultrasound Assessment of the Thyroid Gland in Cattle"

_vetsci, 2023, doi:10.3390/vetsci10050322_

Round 1

Reviewer 1 Report

Dear authors.

although interessting, for me it is really minimal when comparing  9 estimations of 3 persons for technical validation. can you please add a sample size estimation??

Based on what do we now that these estmations are of normal sound calves?

Minor points:

line 30, 31: why adding that 2 of you are diplomate, not usual

line 94: BCS of 2.85 is very accurate isn't it, this might be better presented as 3- 

line 102-103, see before

line 114, ... succesful,  this needs a ref.

in this part of the manuscript: measure and measurement should be replaced by estimate and estimation resp.

line 159: Thyroid ... Figure 1. is Material and methods

line174 in the past time

line 200: the same as line 174

line 201: ref at the end of the sentence

line 208-210: the same as line 201

line 230: to avoid relevant or significant differences

line 256: estimating

Author Response

Dear reviewer,

Please find the answers in the attached file,

Best regards,

Reviewer 2 Report

Dear authors

This is a very well written and easy to read manuscript. Well presented, and clinically applicable.

I have no specific comments to make. The study is properly explain and results are well described and justified.

Kind regards

Author Response

Dear reviewer, thank you very much for this comment.

Best regards

Reviewer 3 Report

This work presents a novel ultrasound exam for thyroid gland in bovines. The work is well presented and then protocol is correctly described. My main concern of the work is related to the sample size, quantification protocol and statistical analysis. Implementing a new imaging technology for applying in a new species requires a bigger sample size than 10 animals, especially after finding differences in the images depending on the body score. I recommend to increase the number of exams and divide the samples in groups depending on the BSC. Another comment is the measuring protocol without confirming the maximum axes. This fact reduces the significance of the work and its conclusions.

Some specific comments:

-          In line 70-71, it would be nice to describe some of the findings they got during the US exam of thyroids in goats. This will give some point of translationally between ruminants.

-          Lines 76-80, same idea as previous. These two papers are the only ones specific to bovines so they should be described in more detail, specifically the ultrasound findings.

-          Line 97, animals used are described as clinically healthy. How do the authors know it? Did they made any examination for including them in the project? Are there any exclusion criteria? This is capital because, as writers said, hypothyroidism can be subclinical and the gland can be anatomically affected without clinical symptoms. I recommend to correlate the ultrasound findings with blood test (TSH and other hormones) for proving the animals are completely healthy.

-          Line 107, authors described using different probes for cows and calves. Because they are running a reproducibility and replicability study, results and measurements should be separated in two groups depending on the probe used for the exam. R&R analysis must be done using the same measurement tool, due to effect on the results.

-          Line 112, please describe the criteria for “most representative image”. Does it mean the biggest diameter? The best tissue contrasts?

-          Line 115, were the authors able to visualize both thyroid lobes in the same Field Of View? How do you confirm the gland symmetry if not?

-          Line 125, How many total measurements did the authors collect? This total number would give the lecturer an idea of the robustness of the project.

-          Figure 1: Image b, please identify the muscles in the superficial level of the image, as it’s done in the image a. Same idea in images c and d.

-          Line 152, this is the one of the weaknesses of the work. As authors realized and honestly noticed, it is a small group of different BCS cows so their results cannot be compared with the others. This is really important for the potential applicability of the technique in bovines with different BCS.

-          Line 163, any explanation for the different width of the right lobe in cows? Maybe it’s an effect of the small sample size?

-          Line 172, any explanation of the strong correlation between examinator 2 and 3 in calves? Maybe because the images of this group have better quality thank the cows group?

-          Line 196, I recommend to include some minimum results from the reference paper for an easy comparison with the authors’ work.

-          Line 217, authors said that the measurements obtained were not done in the maximum axes each time. Is this correct? If so, all their statistics and results are not conclusive, even the R&R results. How can they compare results between examinators when they are not measuring the same thing? If the measurements are not in maximum axes, how do they know they are comparing the same thing between examinators? This point has an extreme relevance in the work. Please clarify.

-          Line 226, authors explained the problematic of quantitative ultrasonography because of the “uncontrolled” parameters that can affect the images (amount of gel, probe pressure, etc…). This is another reason why I strongly recommend to increase the sample size.

-          Line 236, authors’ comment about fat cows is based on a n sample of 2. They should make more scans in animals with the same BCS for giving support to their comments and conclusions.

Author Response

(The authors gave the same response as above.)

Round 2

Reviewer 1 Report

Dear authors,

Thank you very much for the new version and I have seen that you have adapted almost all my suggestions. The addition of ECVDI diplomate and board member of ECBHM is not relevant for this study in my opinion. probably you can adapt that part. No further remarks.

Author Response

Dear reviewer,

Thank you for your feedback. We would very much like to keep the mention of specials in our article, although we understand your opinion. Thank you for your understanding, 

Sincerely

Reviewer 3 Report

Nothing to comment. Almost all the questions were solved.

Author Response

Dear reviewer,

Thank you for your feedback.
Best regards,